# Secure Modern Wireless Communication Network Based on Blockchain Technology

**Radha Raman Chandan** [1] , **Awatef Balobaid** [2] , **Naga Lakshmi Sowjanya Cherukupalli** [3] , **Gururaj H L** [4],*, **Francesco Flammini** [5],* and **Rajesh Natarajan** [6]

1 Department of Computer Science, School of Management Sciences (SMS), Varanasi 221001, India
2 Department of Computer Science, Jazan University, Jazan 45142, Saudi Arabia
3 CSE Department, Koneru Lakshmaiah Education Foundation, Green Fields, Vaddeswaram, Andhra Pradesh, Guntur 522302, India
4 Department of Information Technology, Manipal Institute of Technology Bengaluru, Manipal Academy of Higher Education, Manipal 576104, India
5 IDSIA USI-SUPSI, University of Applied Sciences and Arts of Southern Switzerland, 6928 Manno, Switzerland
6 Information Technology Department, University of Technology and Applied Sciences-Shinas, Shinas 324, Oman
* Correspondence: gururaj.hl@manipal.edu (G.H.L.); francesco.flammini@supsi.ch (F.F.)

**Abstract:** Sixth-generation (6G) wireless networking studies have begun with the global implementation of fifth-generation (5G) wireless systems. It is predicted that multiple heterogeneity applications and facilities may be supported by modern wireless communication networks (MWCNs) with improved effectiveness and protection. Nevertheless, a variety of trust-related problems that are commonly disregarded in network architectures prevent us from achieving this objective. In the current world, MWCN transmits a lot of sensitive information. It is essential to protect MWCN users from harmful attacks and offer them a secure transmission to meet their requirements. A malicious node causes a major attack on reliable data during transmission. Blockchain offers a potential answer for confidentiality and safety as an innovative transformative tool that has emerged in the last few years. Blockchain has been extensively investigated in several domains, including mobile networks and the Internet of Things, as a feasible option for system protection. Therefore, a blockchain-based modal, Transaction Verification Denied conflict with spurious node (TVDCSN) methodology, was presented in this study for wireless communication technologies to detect malicious nodes and prevent attacks. In the suggested mode, malicious nodes will be found and removed from the MWCN and intrusion will be prevented before the sensitive information is transferred to the precise recipient. Detection accuracy, attack prevention, security, network overhead, and computation time are the performance metrics used for evaluation. Various performance measures are used to assess the method's efficacy, and it is compared with more traditional methods.

**Keywords:** blockchain; wireless communication network; malicious node; security protocol; intrusion detection

## 1. Introduction

Over the last several years, the need for contemporary wireless communication networks has increased tremendously. The global deployment of 5G technologies, which has many more capabilities than 4G communications, is approaching. Between 2027 and 2030, the 6G technology, modern wireless communication network architecture with significant AI capability, is anticipated to be introduced into operation. There is an enormous amount of communication as a result of the quick growth of many developing technologies, including artificial intelligence (AI), virtual reality (VR), three-dimensional (3D) media, and the Internet of Everything (IoE). This demonstrates the value of enhancing interaction processes. A civilization with completely autonomous distant administration technologies

is what they are moving toward. MWCN systems are gaining popularity in every aspect of life, including business, medicine, transportation, and space exploration [1]. The following list summarizes MWCN's salient features: An ultra-high-density network is needed to support 5 Gigabyte networking deployments, huge connection, and consistent quality. Small-cell networking has been identified as a key component of MWCN systems, specifically the idea of highly dense small channels. Sensor nodes tend to be placed much closer together in tiny channel networks than they are in other types of ad hoc networks. As a consequence of this, there is often a rather high amount of correlation and redundancy in the data that is perceived by several nodes. This system is also anticipated to ensure the effective utilization of cutting-edge encryption and modulating algorithms, as well as a novel waveform architecture. They will need less expensive network hardware, less expensive deployments, and improved power-saving features in both the networking and consumer device sectors. Almost 80% of mobile congestion is produced indoors. This amount of data can be transferred to indoor densely small cells, freeing up costly and important microcell capabilities. Only a few milliseconds or less will separate the beginning and the completion of the transaction [1].

Wireless signals transfer data at the speed of light in the universe using electromagnetic radiation as transport, which significantly aids in the advancement and growth of the community. At the current time, MWCN's data security problems have drawn a lot of attention as depicted in Figure 1. Anybody within the signal-covering region can eavesdrop on or assault the signal at the physiological layer due to the indigenous "genomic" faults of electromagnetic fields that are exposed by the free transmission of wireless communications. However, current security measures are mostly based on the cryptography method utilized in conventional wired communication and are created to a greater extent, making them unable to effectively address security concerns brought on by the accessibility of communication networks [2]. Blockchain innovation has the prospects to substantially improve the safety of physician and Medicare data technologies that cope with data like patient digital wellness data, medical assent, pharmacy supply chains, blockchain-based remote monitoring records, information for investment businesses, and other confidential material related to scientific experiments. The implementation of blockchain technology can increase medical data transfer efficiency, accessibility, security, and accountability. Blockchain technologies, coupled with artificial intelligence (AI) and machine learning, are about to change the medical industry. The distributed design of blockchain technology is being combined with memory innovation to guarantee the confidentiality of the information for the investors utilizing the public ledger method [3]. Various attack types have occurred during communication between nodes, whether it is within transmission range or beyond the spectrum (i.e., an insider threat or an outcast target). As a result, there are security concerns with forwarding, including data gathering, route maintenance, information propagation, etc. [4]. An unauthorized action or behavior that damages the wireless environment is referred to as an intrusion. In other terms, an intrusion is defined as an attack that compromises the privacy, authenticity, or accessibility of data in any way. Safety threats to the MWCN frequently come from both the inner and outside of the network, where legitimate network nodes can become corrupted and occasionally made to behave maliciously. The timely identification, containment, and elimination of rogue nodes inside a network are other crucial security threats. Addressing security-related challenges has drawn a lot of interest and had a significant influence on MWCN's architecture and evolution patterns [5]. Therefore, we suggested using blockchain-based technologies to safeguard MWCN by detecting malicious nodes and preventing attacks in wireless transmission.

This article is organized as follows: Section 1 describes the introduction, Section 2 examines similar works, Section 3 describes the suggested method, Section 4 presents the results and discussion, and Section 5 provides the conclusion.

**Features of modern wireless communications networks**

**Improved spectrum effectiveness** — **Offloading of High congestion indoors** — **Very minimal latency** — **Ultra-high-density network** — **Low cost** — **Small-cell networks**

**Figure 1.** Features of MWCN.

## 2. Survey of the Literature

The Wireless Multimedia Sensor Network (WNSM) has become more popular among many groups as a result of technical developments in sensors and contemporary gadgets. A network being targeted by many attackers is known as a dispersed assault. Compared to assaults involving a single node, this form of attack greatly worsens network functioning difficulties. An improved machine learning method must be offered to protect the network from the dangers of DoS assaults. An improved Deep Neural Network technique is suggested for WMSN attack detection [6]. A wireless network uses self-organizing modules that are distributed randomly and have a tiny battery capacity to observe the area and allow real activities. Public access is maintained through wireless communication, which encourages a rise in harmful activity inside the network. The majority of network attacks are black hole attacks. In this study, they proposed the Hybrid Deep Learning Prediction (HDLP) framework in the wireless network to maximize battery life and networking reliability [7].

The implementation of fifth-generation (5G) wireless communication technologies was effectively publicized by Non-Orthogonal Multiple Access (NOMA), which is currently regarded as a key innovation in 5G networking. In this study, they created a NOMA model and used a dropping assault to recover a database from the system. Following the use of ML techniques, the retrieved data's detection accuracy for dropping assaults was 95.7%. Additionally, relying on the use of various ML and DL approaches, this study proposes a process for wireless cyber threat identification in 5G technologies [8]. This is the age of smart cognitive radio network innovation, which allows for the effective use of the bandwidth that is now accessible. The goal of cognitive radio innovation should be interference-free frequency availability for consumers. The study addresses various assaults and their causes. The relevance of the authentication system in preventing attacks and ensuring easy frequency use is shown. In this study, the mechanisms and requirements for authentication are examined along with ways to address the safety problems in cognitive networks. The scientific issues surrounding the cognitive radio network's privacy and potential solutions are discussed in this study [9].

Sensors in wireless communication are vulnerable to a variety of security risks. Wireless communication is susceptible to denial-of-service assaults because of these sensors. One of these is a wormhole assault, which alters the network's distribution pathways by using a low-latency connection between two rogue sensor nodes. This assault is harsh because it defies several security techniques and is difficult to detect inside the system. The identification and prevention of wormhole attacks in wireless sensor connectivity is the focus of a thorough assessment of the research in this study [10]. Wireless Sensor Networks (WSNs) are vulnerable to several rogue nodes as a significant information-transmitting technology.

Due to the inadequacy of the current malicious node identification approaches in wireless sensor communications, this research suggested an improved lower energies adaptable clustered hierarch (Enhanced LEACH) routing protocol for harmful node identification based on reputation. A unique method aims to detect rogue nodes in the WSN [11].

The information must be kept secure since it is sent through a wireless channel. The method used in this research helps to ensure that data is safely sent from the origin node to the ground station. This paper proposed a lightweight Bloom filter solution for information transport and packet loss detection in intermediate nodes. They used source data to help them find any malicious packet-discarding nodes and relied on attribution encryption and decryption methods for the model. In this, the data might be discarded while being transmitted [12]. Nodes are vulnerable to several risks as a result of their transparency, among them being deceptive suggestion attacks that provide misleading trust levels that benefit the perpetrator. The artificial bee colony algorithm (ABC) and a fuzzy trust model (FTM-ABC) are utilized in this study to propose a method for identifying malicious nodes. The fuzzy trust model (FTM) is introduced to calculate indirect trust, and the ABC technique is utilized to enhance the trust model to detect false-positive suggestion attacks [13]. In the Malicious Nodes Detection (MND) stage, the Improved Deep Convolutional Neural Network (IDCNN) locates the MN and separates those into the malicious listed box. The Extended K-Means (EKM) method groups the Trusted Nodes (TN) in the energy-efficient stage, and the t-Distribution based Satin Bowerbird Optimization (t-DSBO) method chooses a unique cluster head for every cluster centered on the remaining power of those networks [14].

Malicious assaults (such as wormhole and blackhole assaults) have become a severe problem in wireless transmission in the latest days. Wormhole and blackhole attacks use up more computer resources, network activity, and power. In this study, a brand-new Cross-layer-based Hidden Marko model (C-HMM) is suggested to identify and isolate blackhole and wormhole assaults in wireless ad hoc networks with high efficiency and low transmission costs [14]. The development of wireless communication has only been beneficial to people. In these, data is exchanged between the nodes via the wireless connection at an extremely fast pace. However, one difficulty associated with communication is maintaining confidentiality. They must guarantee that the information packets are transferred privately to the recipient without being accessed by a third party. We provide a technique that uses a node's spatial data attribute to estimate received signal strength (RSS), which is the primary variable for visualizing aggressor nodes in the system and removing the assailant nodes by using clustering methods using a radar grid [15]. This study presents a mechanism for identifying malicious nodes that will make wireless sensor networks much more trustworthy and secure, called density-based spatial clustering of applications with noise (DBSCAN). The major objective of this approach is to design a routing strategy that can detect malicious nodes, has a stronger consistency over time, and has a longer network lifespan. Density-based clustering is a popular and often-used method in many domains. The DBSCAN is a highly popular and effective density-based clustering method that can find clusters of any kind. However, it was unable to identify every node in a network [16]. Numerous drawbacks in the above system, such as low detection accuracy, more energy consumption, and attack prevention, are not effective in wireless communication. Ref. [17] discussed cutting-edge multi-tier authentication techniques that have been presented over the years from 2011 to 2018, their flaws and security concerns, and eventually their solutions for fog computing environments. We compared the various multi-tier authentication solutions based on three criteria: deployment costs, security, and usability. Ref. [18] addressed the multi-stakeholder problem in a fog-enabled cloud. This study proposes a Privacy-Aware Log-preservation Architecture in Fog (PLAF), a comprehensive and automated architecture for proactive forensics in the Internet of Things (IoT). It takes into account the preservation of distributed edge node logs while also being security- and privacy-aware. As previously said, we have created a test bed to implement the specification by combining numerous cutting-edge technologies in one location.

*Problem Statement*

Modern wireless communication networks (MWCN) serve as a crucial means of information transmission. Because everyone inside a wireless network's service region can seek to penetrate the system, wireless networks have insufficient privacy protection. Destructive cyber-attacks have been recorded regularly at locations with accessible, connected networks, and it has been noted that these locations are most susceptible to a total hack of the smartphone or computer data. They might be attacked by several malicious nodes. It is important to eliminate these MWCN inefficiencies. This research presented a blockchain-based mode, Transaction Verification Denied conflict with spurious node (TVDCSN) technique, in light of the ineffectiveness of the conventional malicious node identification and attack prevention approaches in wireless communication networks.

## 3. Research Method

Contemporary technologies have advanced technologically, which has increased interest in the MWCN among diverse populations. Although, because of its wide connectivity it faces several security dangers, one of the main problems for network administrators is authenticating communications in MWCN. Each network layer may be the target of several threats. Even though it would be ideal to provide MWCN with enhanced security measures that can identify network intruders and suggest such remedies, we presented the Transaction Verification Denied conflict with a spurious node to provide secure transmission of sensitive information.

*3.1. Dataset*

Healthcare documents, social media data, and sensor data make up the suggested system's database. Wearable biological and cognitive sensors are used to retrieve the patient's sensory data. People with hyperglycemia and high blood pressure have many variables detected using devices and smart devices. The majority of the signs of diabetes, high blood pressure, and other disorders are covered by the sensed variables. Additional data are also taken out of the person's body. Hospital documents provide information on the therapies that individuals with hypertension and high blood pressure received. They gather patients' health history, which details their health information (including procedures, blood tests, and medication use). This includes the whole patient file in a digital file. This also includes various health information about the patient's condition, including results from testing, responses to questions about one's well-being, and drugs used. A patient's medical state may be evaluated using lab test results from healthcare equipment in the perspective of standards [19].

The content of patients is retrieved from hospital social networking platforms as the initial step of the proposed solution. Nevertheless, further effort is required for this activity, and its success is entirely dependent on the privacy settings of social networking sites.

The application programming interfaces (APIs) of certain social networks are hidden from public view. In a circumstance such as this one, specialized software, such as wrappers, can be utilized to retrieve information (for example, patient posts) [20]. People with diabetes and high blood pressure typically maintain regular contact with their physicians; however, patients with these conditions also require assistance, information, and abilities to personally monitor their healthcare situation. In addition, if patients do not receive useful information from their doctors, social media may be able to perform an important role in satisfying their requirements. As a result, patients can make use of chances provided by social networking platforms such as Facebook and Twitter to acquire sufficient knowledge regarding diabetes and BP and to interact with people who have similar health problems and have had comparable experiences. Patients and medical professionals alike can benefit from the platform that social networks offer for the exchange of information regarding diabetes therapies. To improve patient care and knowledge, we collect data from social media, such as drug reviews and emotional posts made by patients. This allows us to

predict the patients' levels of stress and depression, identify the side effects of diabetes medications on diet and lifestyle, and improve patient care.

The data that make up the system that is being suggested include medical records, sensing data, and data from social networking sites. However, due to its inconsistencies, missing information, noise, multiple formats, vast size, and high complexity, real-world big data is notoriously difficult to work with. The results produced by low-quality and noisy data are also of low quality. The phase of preprocessing the data is performed before the processing itself, which both enhances the overall quality of the processing and reduces the amount of time it takes. The pre-analysis of sensor data, preprocessing and filtering of sensor data, preprocessing of medical records, and preparation of sensor data are all components of our system.

### 3.2. Transaction Verification Denied Conflict with Spurious Node (TVDCSN)

Every node in the suggested technique must only utilize the data that is readily accessible to it, without depending on a centrally or localized trustworthy source. This method examines the validity of the WELCOME information rather than constantly verifying it by searching for inconsistencies between the information and the known architecture. This allows for single *MPR* nominations as long as there are no inconsistencies. An *MPR* may be chosen for any two-hop residents for whom it is the only access point, despite any inconsistencies. However, it cannot be proposed as the exclusive *MPR* for two-hop neighbors that are accessible by other routes.

The notations utilized in the technique are as follows:

$N$ denotes the group of all nodes in the network; the victim and attacking nodes are denoted by $v$, a; $S_y$ is a spurious node that y promotes; the collection of all $v$'s 1-hop neighbors is represented by $HN(v) \subset N$; $HN2(v) \subset HN(v)$ is the collection of all the $v$'s two-hop neighbors; the collection of one-hop nodes of v that designated $v$ as their MPR is known as $MPR(v) \subseteq adi(v)$; and the collection of one-hop nodes chosen by $v$ to serve as *MPR*s is denoted by $MPR'(v) \subseteq HN(v)$.

### 3.2.1. Conflict Rules

We outline the conditions that should be achieved for a node to recognize the sender of a WELCOME text. Take into account $HN(v)$ = b, c, $x$ and $NH2(v)$ = d, e. Depending on the protocol, $v$ must choose $MPR(v)$ = b, c to encompass $HN2(v)$. Assuming that x wants to isolate victim $v$, $y$ sends a false Welcome text with the following contents: $HN(x)$ = v, d, e, $S_y$. The Laws are:

- If node $x$ broadcasts a WELCOME message with $HN(y)$, node v must verify that none of the nodes indicated by $x$ are one of $HN(v)$. Nodes b and c must be present in $HN2(y)$; therefore, $y$ should choose *MPR*s that would enable it to connect to them. Nevertheless, $y$ may pretend to wish to select $v$ as *MPR* for taking care of a and b;
- If a node $y$ is named in a Welcome text, node $v$ must check to determine if there is a node u $HN(y)$ that is (a) not referenced in the recipient's WELCOME text and (b) at least three hops distant from node $v$. If such criteria are met, a secondary assessment is required: (c) has w been designated as *MPR* to fill in for $u$ by $y$? Figure 2 shows the finding of conflicts to prevent attacks.
- Accessing the Topology management (*TM*) table might be used to perform assessments (a) and (b). There is a conflict if there isn't an element carrying the *MPR* that $x$ selected and that enables it to go from $x$ to $z$ in just two hops. Keep in mind that if either condition (a) or (b) is not met, conflicts cannot be found. A *TM* text must exist where either $y$ has chosen $u$ or $u$ has chosen $x$ as *MPR* for (c) to be verified. To do this, $v$ must check each u 2 $U$, where $U$ $HN2(y)$ is dependent on $y$'s text. Algorithm 1 illustrates the testing of the criterion when the *TM* message's structure is "latest (location), dest (location)";
- A WELCOME text with all $Hn(v)$ must be viewed by $v$ as a threat, and necessary action must be taken.

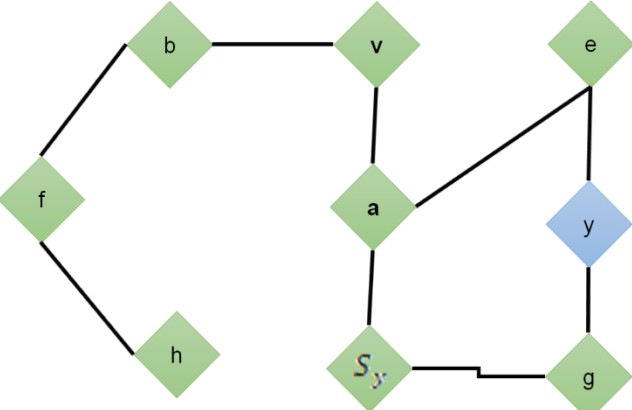

**Figure 2.** Detecting conflicts.

---

**Algorithm 1: Testing criterion**

---

Testing–criterion $(TM,\ H,\ X,\ V)$
$U \leftarrow \Phi$
   For each $r \epsilon TM$ do
      If r.last $\epsilon\ HN(y)$ do
 $U \leftarrow U\ Z\ \{r, dest\}$
      If r.dest $\epsilon\ HN(x)$ do
         $U \leftarrow U\ Z\ \{r, last\}$
   For each $u\ \epsilon\ U$ do
      If $u\ \epsilon\ U\ \cap\ HN(v)$ do
 $U \leftarrow U - \{u\}$
   For each $m\ \epsilon\ MPR'(x)$ do
   For each $U \epsilon U$ do
If $\{m, y\} \epsilon\ TC$ such that z is encompassed by m do
 $U \leftarrow U - \{u\}$
$if\ U \leftarrow \Phi\ do$
   Consider $y$ as a malicious node
Else
Consider $y$ as a trustworthy *MPR*

---

Using a spurious node, this looks for discrepancies between a WELCOME signal and the system architecture as it is known from previous WELCOME and *TM* messages. However, make sure to double-check each node that the WELCOME message mentions. There are situations in which a node isolation assault is still possible. Think about Figure 3, where y falsely claims that $HN(y)$ = v, f, e, and g. *MPR* (y) = "f, h" and $HN2(x)$ = "a, b, e, j, l". There are no contradictions that v can find because y does not assert that it is aware of any node in $HN(v)$ except itself (rule No. 1). a, b, e, j, and l are the *MPRs* that were chosen by $y$ to access all of $HN2(y)$. Since d is previously approachable by f (rule No. 2) and $y$ does not claim to be aware of all of $HN\ (v)$, in particular b, it is predicted that $x$ would not designate c as one of its *MPRs* (rule No. 3).

Regrettably, if each node in the system declared an extra fake node, all nodes would be recognized as *MPRs* as a result of their false advertisements, and the network would return to Link-State Forwarding. As a result, a technique for restricting false messages must be developed that finds a balance between the requirement to minimize node usage and preserving the network against separation assault.

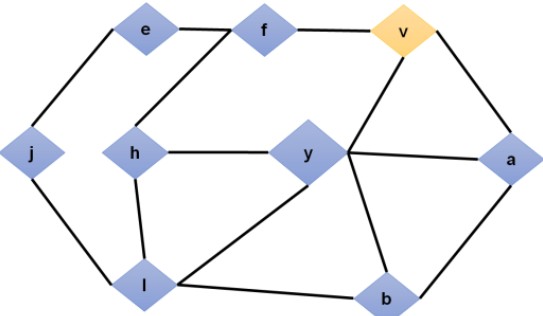

**Figure 3.** Node attack with no conflicts.

To avoid nodes in the networking from informing the others of misleading data about their connection, we built up a method enabling each node to determine if an attack may be launched via itself. If such a falsehood is feasible, the node creates a spurious node and connects it to the network to stop others from believing they are connected to it. In other words, the nodes themselves are in charge of ensuring that the connection data is accurate since they should prevent others from misusing it. The following provides the limiting method for introducing or eliminating spurious nodes:

- When all nodes in *HN2* (*v*) ∈ *HN* (*v*) imply that the separation between *y* and *u* is less than three-hops, every node v must incorporate a spurious node;
- *Sv* ∈*HN*(*v*);
- New node *u* promotes Fu by nature before rule 1 is calculated;
- Then, the spurious node is removed when rule (1) is falsifiable;
- Regular inspection must be carried out (every spurious verification period).

There are no nodes in Figure 4 with a separation equal to 3 from any of the nodes {*y*, j} ∈ *HN2*(b). As a result, node c should add a spurious node to the system following rule No. 1 of the fake setting method. Because node *y* should designate b as an *MPR* to approach *Sv*, this prevents the assault and safeguards node *v*. This would be reported as a conflict and in violation of rule No. 2 of the conflict rules. Through this method, the attacks can be prevented, and malicious nodes are identified.

The trust levels of every node in a system, including malevolent nodes, are updated by block transactions. A block will be created by the validating node or a delegation node, which receives all activities. Transactions are distributed by *MPR* nodes under the mechanism used by this method. Every node n will deliver an encoded session (n, transaction) prKeyn, where the secret key of n is used to encode the operation. If the abovementioned process reveals a malevolent node, it will be given a low Trust value (TV) and removed from the system. Even though a node is not an enemy, one node could mistakenly attribute a negative rating to it. Transactions including malevolent node data must first be verified by neighbors before being forwarded to the delegation node to avoid this problem. Because hackers may assert that two neighborhoods of a target are their counterparts in a node attack (NA), the intruder's data and any discrepancies they create must be notified by two neighborhoods. The suspect's secret key is used to encode the target ID (*v*), assailant ID (*y*), and Reporting Attempt (discovered discrepancies) in a response signal (*v*, *y*, Report Attack) prKeyv that is transmitted. Because the malignant welcome data contain the suspect's two-hop neighborhood, this signal is delivered by piggybacking onto it until it is within two hops of the recipient.

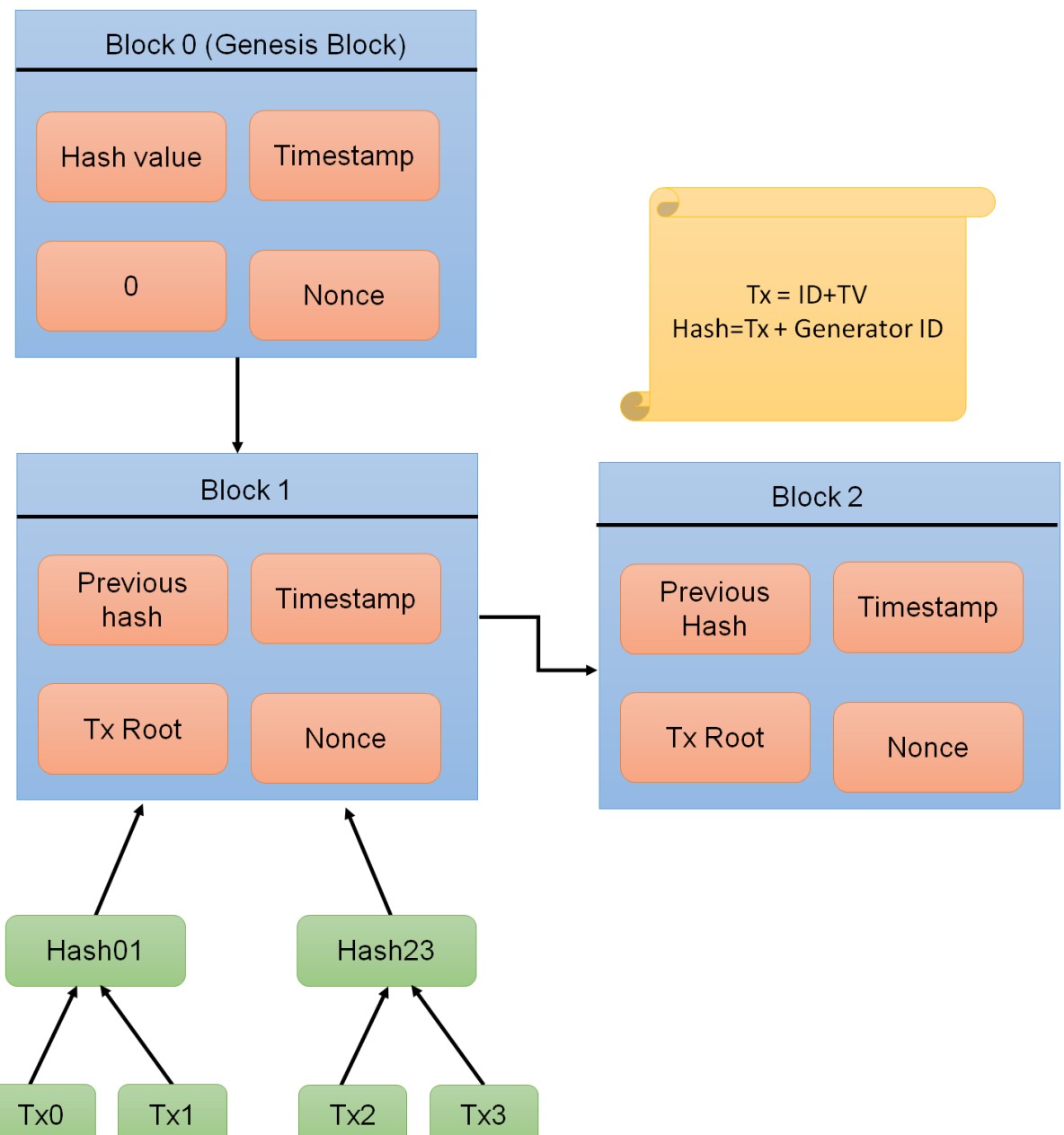

**Figure 4.** Sample Block Configurations.

If the surrounding nodes accept the transaction, it will respond (i AckReport) prKeyi, validating the transaction. It is hard to receive consensus from all endpoints since the hacker might also incorporate the spurious nodes. Furthermore, the node asking for permission can alternatively be an intruder attempting to identify a reliable node. As a result, the transaction is approved if at least half of the neighbors who received the intruder's Welcome approve. Additionally, even if the intruder states that they are its neighborhood, saying "accept" suggests that they have no link to them. As a result, each node evaluates whether or not they agree using the same criterion. Nodes that have TVs greater than q are the only ones that can transmit non-attacking standard TV transactions. The delegation node will tally the nodes participating in a specific transaction's vote. The delegation will choose the transaction order and create a block depending on the quantity. As a result, using *MPR* nodes across the network, the delegates will disseminate the new block (dl, Block) prKeydl. Every node responds with a verification signal (n, BlockAck) prKeyn after receiving the

block from all other nodes. Every node links the new block to a localized blockchain if the most of other nodes approve it. Through this procedure, the suggested method will identify the malicious node and eliminate the node and its attack in the MWCN transmission.

### 3.2.2. Block Configuration

When building a block, it is important to specify the data that will be contained within it as well as how the delegate node will configure it. In a blockchain system, the pool's transactions are compiled into a block and chained throughout the network because it offers immutability. A hash value (SHA-256 algorithm) is attached to the block in a blockchain and is directly derived from the transaction data. As a result, the hash value will alter even a minor modification in the data. A data update in one block might cause all the other blocks in a blockchain to become disorganized since the hash of the previous block will be incorporated as data in the current block for chaining. There is only one format that the block hash accepts (e.g., a hash signature starting with 10 consecutive zeros). The term "nonce" refers to a piece of data that complies with this criteria. Until a valid hash signature is obtained, the nonce value is continuously modified.

Blocks in a MANET trust blockchain are made up of block transaction data and the aforementioned metadata (timestamp, hash of the transaction, delegate ID, and the nonce). To ensure non-repudiation for the block transactions offered by any nodes, the transaction generator ID, the TVs recommended by the transaction generator, and the delegate ID will all be included when a transaction is hashed.

The first block in the blockchain, known as a "genesis block" (blockchain jargon), is defined as an empty list of transactions when the network is created. Figure 4 displays a sample arrangement for a block.

### 3.2.3. Block Maintenance

There are two sorts of nodes in a blockchain environment: full nodes, which maintain the blockchain, and lite nodes, which mostly rely on full nodes for information but do not maintain the whole blockchain. We included this idea in our environment as well by the nature of MANETs. A new node will have access to the blockchain data whenever it joins the network. As seen in Figure 2, a node should initially join the network as a light node, which allows it to only download the block's header. A new node can nevertheless produce transactions (attacker detection/TV calculation) in the network even though it will initially function as a light node. To relay block headers until the new node becomes a full node, the network's host node will act as a temporary full node.

## 4. Results and Discussion

This section displays the findings of the graphical assessments of the efficacy of the suggested and existing strategies. Using the suggested TVDCSN approach, malicious node elimination and intrusion avoidance are carried out. The performance indicators for evaluation include detection accuracy, attack prevention, security, network overhead, and computation time. The suggested TVDCSN is used to compare the performance of the Transfer learning (TL), AdaBoost Regression Classifier (ABRC), malicious intrusion data mining algorithm (MIDTA), and dynamic reputation algorithm (DRA).

### 4.1. Detection Accuracy (%)

Accurately identifying malicious nodes in a wireless communication network is the definition of detection accuracy. The malicious node will reduce the network's communication speed, which would reduce the network's service time. It is necessary to identify these wireless communication nodes. Figure 5 displays the detection accuracy of malicious nodes using both existing and suggested methods. It shows that the proposed approach is effective in detecting precise malicious nodes. Table 1 displays the results for the detection accuracy.

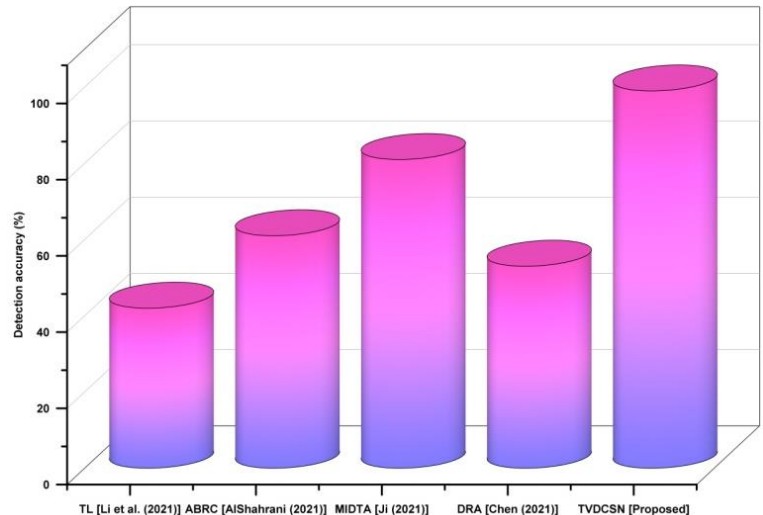

**Figure 5.** Proposed and existing methods of detection accuracy.

**Table 1.** Values of proposed and existing methods of detection accuracy.

| Methods | Detection Accuracy (%) |
|---|---|
| TL (Li et al. (2021)) | 42 |
| ABRC (AlShahrani (2021)) | 61 |
| MIDTA (Ji (2021)) | 81 |
| DRA (Chen (2021)) | 53 |
| TVDCSN (Proposed) | 99 |

*4.2. Attack Prevention (%)*

During the process of transmitting sensitive information through the MWCN, the network is subject to several attacks. Numerous vulnerable attackers that want to steal sensitive information are the ones who carry out these attacks. In the transmission process, the prevention of attacks is vital. The attack prevention employing both recommended and existing approaches is shown in Figure 6. The attack prevention results are shown in Table 2. It demonstrates how well the suggested strategy works to prevent attacks in MWCN.

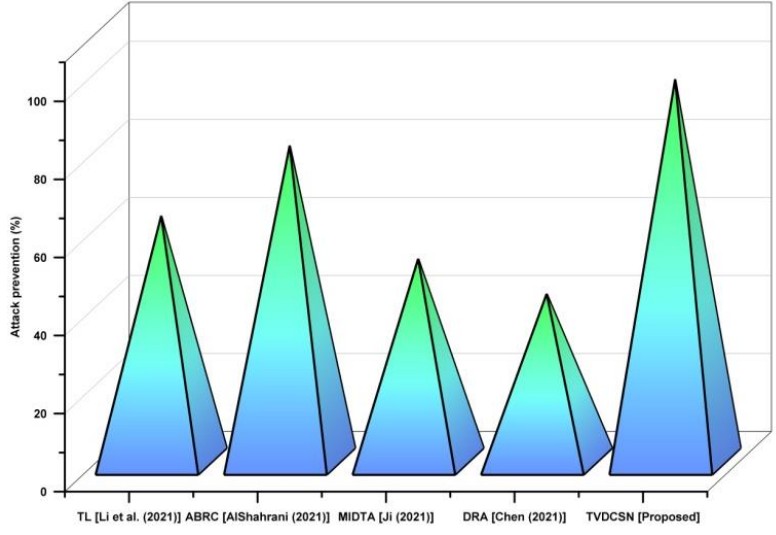

**Figure 6.** Proposed and existing methods of attack prevention.

**Table 2.** Values of proposed and existing methods of attack prevention.

| Methods | Attack Prevention (%) |
|---|---|
| TL (Li et al. (2021)) | 63 |
| ABRC (AlShahrani (2021)) | 81 |
| MIDTA (Ji (2021)) | 52 |
| DRA (Chen (2021)) | 43 |
| TVDCSN (Proposed) | 98 |

### 4.3. Security (%)

It is essential to have security because it protects sensitive data from being compromised by malicious cyber activity and ensures that the network can be relied upon and is functional at all times. Various security measures are used in effective network security plans to shield people and companies from ransomware and digital threats. Figure 7 shows the security utilizing both the recommended and existing techniques. This demonstrates that the strategy that was proposed is an effective one for providing security. The outcomes for the security are shown in Table 3.

The formula for network security NS = P + Pr + Pe + M + T; NS—Network security, P—policy, Pr—procedure, Pe—people, M—management, and T—technology. The effective collection of data to test and evaluate situational awareness and treat assessment tools for cyber security will be made possible by this adaptable simulation modeling framework.

### 4.4. Network Overhead (Bits)

Any unlawful use of services such as data, processing, storage, and bandwidth is referred to as network overhead in computing. To hold the additional data required to transport specific information from its source to its recipient, more assets are required. Figure 8 depicts the network overhead of the suggested and current strategies. It shows that the recommended solution has minimal overhead, which enhances the wireless communication network. Table 4 displays the overhead values. The below equation illustrates the comparison of network overhead.

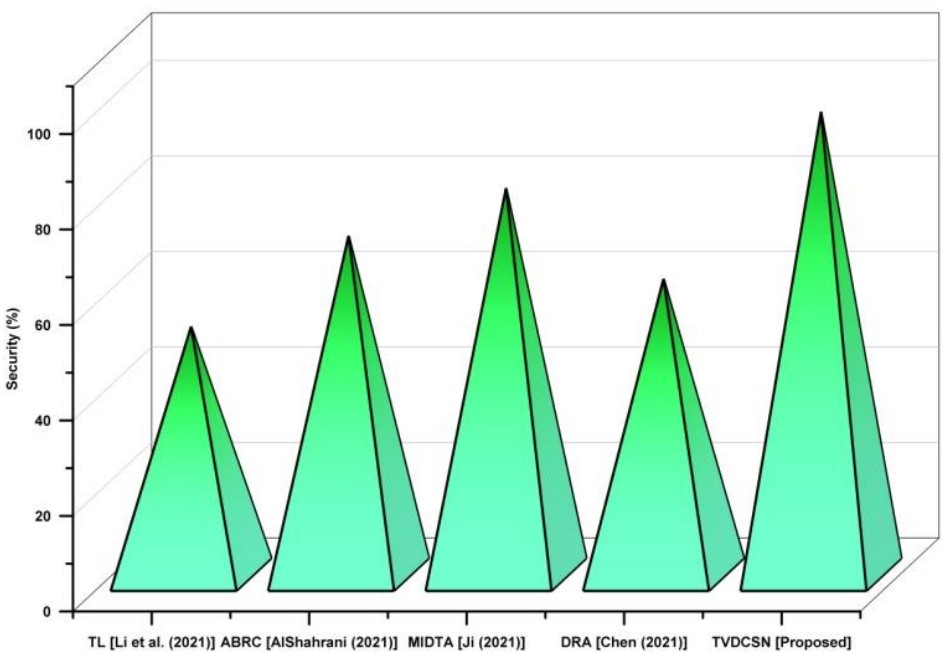

**Figure 7.** Proposed and existing methods of security.

**Table 3.** Values of proposed and existing methods of security.

| Methods | Security (%) |
|---|---|
| TL (Li et al. (2021)) | 52 |
| ABRC (AlShahrani (2021)) | 71 |
| MIDTA (Ji (2021)) | 81 |
| DRA (Chen (2021)) | 62 |
| TVDCSN (Proposed) | 97 |

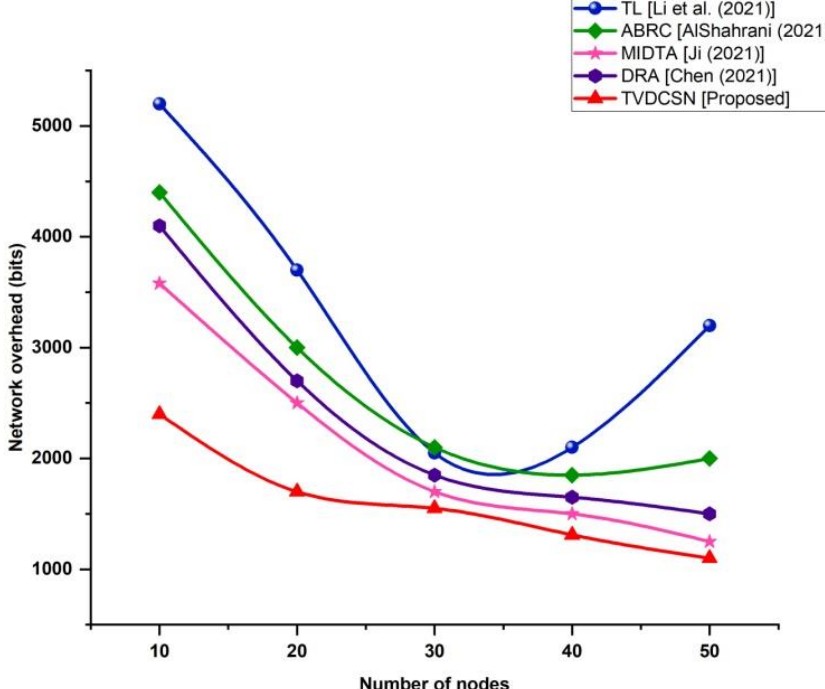

**Figure 8.** Proposed and existing methods of network overhead.

**Table 4.** Values of proposed and existing methods network overhead.

| Number of Nodes | Network Overhead (Bits) | | | | |
|---|---|---|---|---|---|
| | TL (Li et al. (2021)) | ABRC (AlShahrani (2021)) | MIDTA (Ji (2021)) | DRA (Chen (2021)) | TVDCSN (Proposed) |
| 10 | 5200 | 4400 | 3580 | 4100 | 2400 |
| 20 | 3700 | 3000 | 2500 | 2700 | 1700 |
| 30 | 2050 | 2100 | 1700 | 1850 | 1550 |
| 40 | 2100 | 1850 | 1500 | 1650 | 1310 |
| 50 | 3200 | 2000 | 1250 | 1500 | 1100 |

In that situation, O = 2l, where l is the number of connections that calculates the number of overhead networks, O.

### 4.5. Computation Time (%)

Computation is the amount of time required to accomplish a calculation (also known as "execution periods"). It is a fundamental efficiency criterion that professionals in the fields of software engineering and science have used to evaluate a method's effectiveness. Figure 9 displays the computation times for the suggested and traditional methodologies. Table 5 displays the values of calculation time. It indicates that the suggested strategy operates effectively and rapidly.

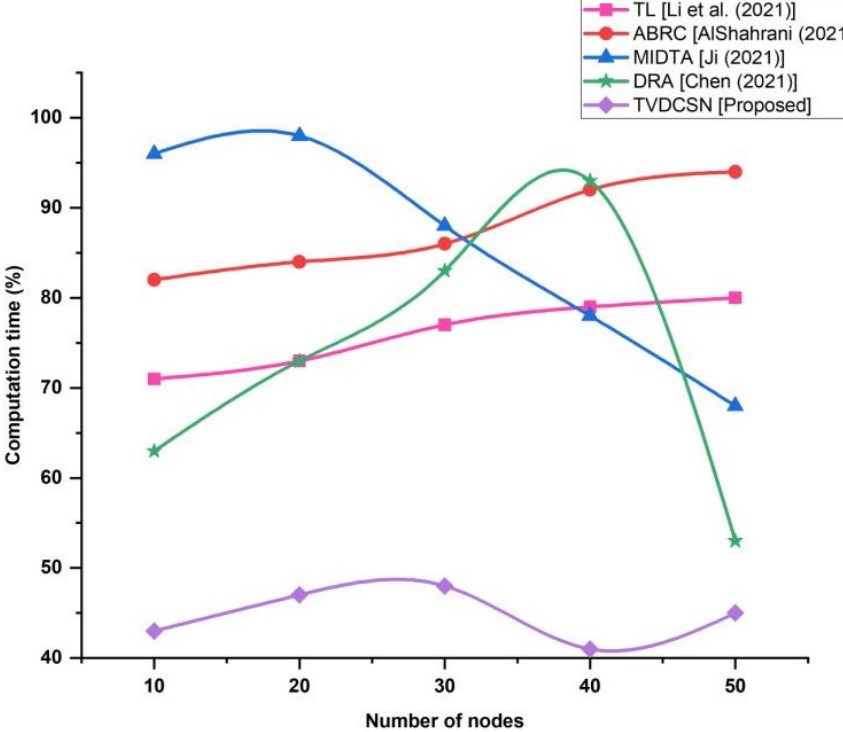

**Figure 9.** Proposed and existing methods of computation time.

*4.6. Block Latency*

Block latency is improved even more if an attack detector node serves as the delegate since less communication is needed to send attack information to the delegate. The block duration and transaction ratio are significantly lower when collusive attacks take place in a network. The block generation latency, determined based on attack transactions, can be depicted in Figure 10.

**Table 5.** Values of proposed and existing methods computation time.

| Number of Nodes | Computation Time (%) | | | | |
|---|---|---|---|---|---|
| | TL (Li et al. (2021)) | ABRC (AlShahrani (2021)) | MIDTA (Ji (2021)) | DRA (Chen (2021)) | TVDCSN (Proposed) |
| 10 | 71 | 82 | 96 | 63 | 43 |
| 20 | 73 | 84 | 98 | 73 | 47 |
| 30 | 77 | 86 | 88 | 83 | 48 |
| 40 | 79 | 92 | 78 | 93 | 41 |
| 50 | 80 | 94 | 68 | 53 | 45 |

$$Attack\_ratio = \frac{number\_of\_attacker\_in\_the\_network}{number\_of\_nodes\_in\_the\_netowrk} \tag{1}$$

For instance, if two different attackers initiate attacks simultaneously in two different locations, two assault transactions will be included in a block, increasing the effectiveness of the suggested technique. The attack ratio measurement is shown in Equation (1).

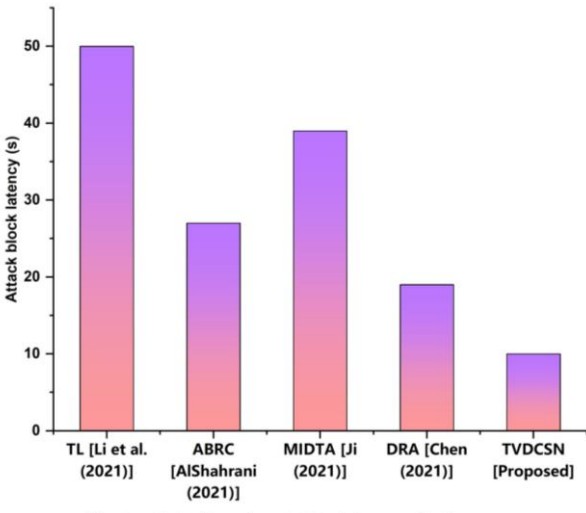

**Figure 10.** Block Generation Latency Based on Attack Transactions.

## 5. Discussion

In the area of WCN data analysis, wireless transmission system assault detection is crucial. In unsupervised wireless systems, link forecasting is a challenging issue that can be effectively handled by transfer learning (TL). A link prediction approach relying on the dispersion functional fitting technique of the area diagonal term group is utilized to gather more precise and comprehensive data in the target area [21]. Several security failures have occurred lately as a result of the unfavorable growth of automation. Services are maximized with increased network lifespans to resist those safety dangers and intrusions, particularly for hacking attempts. Artificial intelligence depended on innovation and has advanced to resist intrusions. Deep learning (DL) depended on a categorization strategy for detecting cyber-attacks provided in this article [22,23]. The intrusion detection system using the suggested AdaBoost Regression Classifier (ABRC) uses a deep learning structure. The presented ABRC with DL architecture is implicated in the assessment of network security assault. The privacy of private details in wireless technology cannot be guaranteed because invasive information in the transmission process readily affects wireless private interaction networks. The malicious intrusion data mining algorithm (MIDMA) presented in this study [24,25] is founded on valid large information from wireless personal interaction systems. The main point of malicious infiltration data is repeatedly obtained using the grouping technique, and its predicted participation is determined. The inherent complexity of wireless communication networks makes it difficult to identify rogue nodes using standard approaches, which creates several safety threats in the network setting. In this research, a dynamic reputation algorithm-based technique for detecting rogue wireless transmission nodes is proposed [26,27]. The above methods take a long time to identify and detect malicious activity with less accuracy and fail to effectively prevent attacks.

## 6. Conclusions

In a "wireless communication network" where the communication of information is fully automated by utilizing electromagnetic waves, like radio waves, which are typically instituted in the physical layer of the system, one of the most significant methods for transferring data between nodes without utilizing wires is used. In the area of data transfer, wireless communication systems have made significant progress to date. This is because they are easy to operate, affordable, and have sufficient bandwidth. The security risks to wirelessly transferred data have risen even if the safety and bandwidth gaps between different kinds of networks have decreased as a result of ongoing advancements in wireless communication innovation. The MWCN has to take measures to reduce the number of

security-related issues. As a result, we offered the blockchain-based modal, Transaction Verification Denied conflict with spurious node (TVDCSN) methodology, to be used in MWCN because of the inefficiency of the traditional methods for identifying malicious nodes and preventing attacks. The efficacy of the proposed system is assessed using a variety of performance characteristics, including detection accuracy, attack prevention, security, network overhead, computation time, and average block latency. The proposed method's efficacy is compared with that of conventional techniques such as Transfer learning (TL), AdaBoost Regression Classifier (ABRC), Malicious Intrusion Data Mining Algorithm (MIDTA), and Dynamic Reputation Algorithm (DRA). These assessment results demonstrate the effectiveness of the suggested approach in MWCN for detecting malicious nodes and preventing attacks. Even if an attacker moves around and attacks different nodes from different places, the network will still be safe. No information or time is lost, and the overall level of complexity goes down. Additionally, because of collaborative detection, each node is much less responsible for its actions. The more nodes there are in a network, the less each one is responsible for detecting. In the future, optimization strategies may be introduced into the system to enhance its performance. The proposed scheme will be put to the test with different routing protocols in a wireless communication network.

**Author Contributions:** Conceptualization, R.R.C.; Methodology, R.R.C.; Software, N.L.S.C.; Validation, N.L.S.C.; Investigation, N.L.S.C.; Resources, A.B.; Data curation, A.B.; Writing—original draft, G.H.L. and F.F.; Writing—review & editing, G.H.L., F.F. and R.N.; Supervision, R.N. All authors have read and agreed to the published version of the manuscript.

**Funding:** This research received no external funding.

**Conflicts of Interest:** The author declares no conflict of interest.

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
