# Peer review of "Secure Modern Wireless Communication Network Based on Blockchain Technology"

_electronics, doi:10.3390/electronics12051095_

Round 1
Reviewer 1 Report
This paper presents a new methodology for detecting and preventing malicious nodes in mobile wireless communication networks (MWCN) using the blockchain-based Transaction Verification Denied conflict with spurious node (TVDCSN) approach. The authors evaluate the performance of the proposed method by comparing it to conventional techniques such as Transfer learning (TL), AdaBoost Regression Classifier (ABRC), Malicious Intrusion Data Mining Algorithm (MIDTA), and Dynamic Reputation Algorithm (DRA) in terms of various performance characteristics such as detection accuracy, attack prevention, security, network overhead, and computation time.
Overall, the paper provides a comprehensive overview of the current state-of-the-art in MWCN security and the limitations of traditional methods. The authors' contribution of the blockchain-based TVDCSN method is a significant improvement in the field and provides a new approach for detecting and preventing malicious nodes in MWCN.
However, there are several areas where the paper can be improved:
-
Methodology: The methodology section could be expanded to provide more detail on the proposed TVDCSN method, including the algorithms and data structures used, the process for detecting malicious nodes, and the procedures for preventing attacks.
-
Evaluation: The evaluation of the proposed method could be more comprehensive and systematic, including a more rigorous comparison with the conventional techniques, and a larger dataset for testing.
-
Limitations: The paper does not fully address the limitations and challenges of the proposed method. A discussion of these limitations and how they can be overcome in future work would strengthen the paper.
-
Future Work: The conclusion section could be expanded to include a discussion of future work, including potential optimizations and directions for further research in the field.
- The references should be updated by citing up to date papers. Some papers are provided as examples: i. https://onlinelibrary.wiley.com/doi/full/10.1002/dac.4033; ii. https://www.mdpi.com/2079-9292/9/7/1172
In conclusion, the paper provides a valuable contribution to the field of MWCN security, but there is room for improvement in terms of methodology, evaluation, and discussion of limitations and future work.
Author Response
Reviewer-1
This paper presents a new methodology for detecting and preventing malicious nodes in mobile wireless communication networks (MWCN) using the blockchain-based Transaction Verification Denied conflict with spurious node (TVDCSN) approach. The authors evaluate the performance of the proposed method by comparing it to conventional techniques such as Transfer learning (TL), AdaBoost Regression Classifier (ABRC), Malicious Intrusion Data Mining Algorithm (MIDTA), and Dynamic Reputation Algorithm (DRA) in terms of various performance characteristics such as detection accuracy, attack prevention, security, network overhead, and computation time.
Overall, the paper provides a comprehensive overview of the current state-of-the-art in MWCN security and the limitations of traditional methods. The authors' contribution of the blockchain-based TVDCSN method is a significant improvement in the field and provides a new approach for detecting and preventing malicious nodes in MWCN.
However, there are several areas where the paper can be improved:
- Methodology: The methodology section could be expanded to provide more detail on the proposed TVDCSN method, including the algorithms and data structures used, the process for detecting malicious nodes, and the procedures for preventing attacks.
Answer: Thank you for comments. based on the comment we have updated the manuscript.
4.7 Block configurations
When building a block, it is important to specify the data that will be contained within it as well as how the delegate node will configure it. In a blockchain system, the pool's transactions are compiled into a block and chained throughout the network. Because it offers immutability. A hash value (SHA-256 algorithm) is attached to the block in a blockchain and is directly derived from the transaction data. As a result, the hash value will alter even a minor modification in the data. A data update in one block might cause all the other blocks in a blockchain to become disorganised since the hash of the previous block will be incorporated as data in the current block for chaining. There is only one format that the block hash accepts (e.g., a hash signature starting with 10 consecutive zeros). The term "nonce" refers to a piece of data that complies with these criteria. Until a valid hash signature is obtained, the nonce value is continuously modified. Blocks in a MANET trust blockchain are made up of block transaction data and the aforementioned metadata (timestamp, hash of the transaction, delegate ID, and the nonce). To ensure non-repudiation for the block transactions offered by any nodes, the transaction generator ID, the TVs recommended by the transaction generator, and the delegate ID will all be included when a transaction is hashed.The first block in the blockchain, known as a "genesis block" (blockchain jargon), is defined with an empty list of transactions when the network is created. Figure 4 displays a sample arrangement for a block.
Figure 4 Sample Block Configurations
3.2.3 Block Maintenance
There are two sorts of nodes in a blockchain environment: full nodes, which maintain the blockchain, and lite nodes, which mostly rely on full nodes for information but do not maintain the whole blockchain. We included this idea into our environment as well, in accordance with the nature of MANETs. A new node will have access to the blockchain data whenever it joins the network. As seen in Figure 5, a node should initially join the network as a light node, which allows it to only download the block's header. A new node can nevertheless produce transactions (attacker detection/TV calculation) in the network even though it will initially function as a light node. In order to relay block headers until the new node becomes a full node, the network's host node will act as a temporary full node.
Figure 5 Expression of Phase 4
- Evaluation: The evaluation of the proposed method could be more comprehensive and systematic, including a more rigorous comparison with the conventional techniques, and a larger dataset for testing.
Answer: Thank you for comments. based on the comment we have updated the manuscripts
4.6 Block latency
Block latency is improved even more if an attack detector node serves as the delegate since less communication is needed to send attack information to the delegate. The block duration and transaction ratio are significantly lower when collusive attacks take place in a network.
Figure 9. Block Generation Latency Based on Attack Transactions
(3)
For instance, if two different attackers initiate attacks simultaneously in two different locations, two assault transactions will be included in a block, increasing the effectiveness of the suggested technique. Attack ratio measurement is shown in equation (3).
- Limitations: The paper does not fully address the limitations and challenges of the proposed method. A discussion of these limitations and how they can be overcome in future work would strengthen the paper.
Answer: Thank you for comments. based on the comment we have updated the manuscript.
Even if an attacker moves around and attacks different nodes from different places, the network will still be safe. No information or time is lost, and the overall level of complexity goes down. Also, because of collaborative detection, each node is much less responsible for its own actions. The more nodes there are in a network, the less each one is responsible for detecting
- Future Work: The conclusion section could be expanded to include a discussion of future work, including potential optimizations and directions for further research in the field.
Answer: Thank you for comment. based on the comment, we have updated the manuscript.
In future work, the proposed scheme will be put to the test with different routing protocols in wirelss communication network.
.
- The references should be updated by citing up to date papers. Some papers are provided as examples: i. https://onlinelibrary.wiley.com/doi/full/10.1002/dac.4033; ii. https://www.mdpi.com/2079-9292/9/7/1172
- Answer: Thank you for comment. based on the comment, we have updated the manuscript.
Kouhalvandi et al. (2022) discussed cutting-edge multi-tier authentication techniques that have been presented over the years of 2011 to 2018, their flaws and security concerns, and eventually their solutions for fog computing environments. We compared the various multi-tier authentication solutions based on three criteria: deployment costs, security, and usability. Janjua et al. (2020) addressed the multi-stakeholder problem in a fog-enabled cloud, this study proposes Privacy-Aware Log-preservation Architecture in Fog (PLAF), a comprehensive and automated architecture for proactive forensics in the Internet of Things (IoT). It takes into account the preservation of distributed edge node logs while also being security- and privacy-aware. As previously said, we have created a test-bed to implement the specification by combining numerous cutting-edge technologies in one location.
- In conclusion, the paper provides a valuable contribution to the field of MWCN security, but there is room for improvement in terms of methodology, evaluation, and discussion of limitations and future work.
Answer: Thank you for comment. Based on the comment we have updated the manuscripts.
Even if an attacker moves around and attacks different nodes from different places, the network will still be safe. No information or time is lost, and the overall level of complexity goes down. Also, because of collaborative detection, each node is much less responsible for its own actions. The more nodes there are in a network, the less each one is responsible for detecting. In future work, the proposed scheme will be put to the test with different routing protocols in wireless communication network.

Reviewer 2 Report
1. line 53 "specifically the idea of highly dense small channels" -- what does the authors mean by small channels?
2. Extensive improve to the English language is needed. Please use grammarly or any other tool to get help with it.
3. Can the authors explain the lines 312 - 313 "If such a falsehood is feasible, the node creates a spurious node and connects it to the network to stop others from believing they are connected to it."?
4. The authors should provide more information on the data distribution in the dataset. High accuracy might not always indicate a higher performance depending on the dataset.
5. For the sections 4.3 and 4.4, the authors should define how the security and network overhead values are computed.
Author Response
Reviewer-2
- line 53 "specifically the idea of highly dense small channels" -- what does the authors mean by small channels?
Answer: Thank you for comment. Based on the comment we have updated the manuscript.
Sensor nodes tend to be placed much closer together in tiny channel networks than they are in other types of ad hoc networks. As a consequence of this, there is often a rather high amount of correlation and redundancy in the data that is perceived by several nodes.
- Extensive improve to the English language is needed. Please use grammarly or any other tool to get help with it.
Answer: Thank you for comment. Based on the comment we have updated the manuscript.
- Can the authors explain the lines 312 - 313 "If such a falsehood is feasible, the node creates a spurious node and connects it to the network to stop others from believing they are connected to it."?
Answer: Thank you for comment. Based on the comment we have updated the manuscript.
Yes, it is possible for the node to perpetrate such a deception, it will establish a false node and link it to the network in order to trick the other nodes into thinking that they are not connected to it.
- The authors should provide more information on the data distribution in the dataset. High accuracy might not always indicate a higher performance depending on the dataset.
Answer: Thank you for comment. Based on the comment we have updated the manuscript.
The content of patients is retrieved from hospital social networking platforms as the initial step of the proposed solution. Nevertheless, further effort is required for this activity, and its success is entirely dependent on the privacy settings of social networking sites.
The application programming interfaces (APIs) of certain social networks are hidden from public view. In a circumstance such as this one, specialised software, such as wrappers, can be utilised to retrieve information (for example, patient posts) [28]. Persons with diabetes and high blood pressure typically maintain regular contact with their physicians; however, patients with these conditions also require assistance, information, and abilities in order to personally monitor their healthcare situation. In addition, if patients do not receive useful information from their doctors, social media may be able to perform an important role in satisfying their requirements. As a result, patients can take use of chances provided by social networking platforms such as Facebook and Twitter to acquire sufficient knowledge regarding diabetes and BP and to interact with people who have similar health problems and have had comparable experiences. Patients and medical professionals alike can benefit from the platform that social networks offer for the exchange of information regarding diabetes therapies. In order to improve patient care and knowledge, we collect data from social media, such as drug reviews and emotional posts made by patients. This allows us to predict the patients' levels of stress and depression, identify the side effects of diabetes medications in relation to diet and lifestyle, and improve patient care.
The data that make up the system that is being suggested include medical records, sensing data, and data from social networking sites. However, due to its inconsistencies, missing information, noise, multiple formats, vast size, and high complexity, real-world big data is notoriously difficult to work with. The results produced by low-quality and noisy data are also of low-quality. The phase of preprocessing the data is performed before the processing itself, which both enhances the overall quality of the processing and reduces the amount of time it takes. The pre-analysis of sensor data, preprocessing and filtering of sensor data, preprocessing of medical records, and preparation of sensor data are all components of our system.
- For the sections 4.3 and 4.4, the authors should define how the security and network overhead values are computed.
Answer: Thank you for comment. Based on the comment we have updated the manuscript.
In that situation, O = 2l, where l is the number of connections, calculates the number of overhead networks , O.
The formula for network security
NS=P+Pr+Pe+M+T
NS-Network security-policy, pr-procedure-people-management and T-technology
The effective collection of data to test and evaluate situational awareness and treat assessment tools for cyber security will be made possible by this adaptable simulation modeling framework.

Round 2
Reviewer 1 Report
Thank you for addressing my recommendations. The paper is now suitable for publication.
Reviewer 2 Report
All questions have been answered.